# SYMMETRY LEADS TO STRUCTURED CONSTRAINT OF LEARNING

## ABSTRACT

Due to common architecture designs, symmetries exist extensively in contemporary neural networks. In this work, we unveil the importance of the loss function symmetries in affecting, if not deciding, the learning behavior of machine learning models. We prove that every mirror symmetry of the loss function leads to a structured constraint, which becomes satisfied when either the weight decay or gradient noise is large. As direct corollaries, we show that rescaling symmetry leads to sparsity, rotation symmetry leads to low rankness, and permutation symmetry leads to homogeneous ensembling. Then, we show that the theoretical framework can explain the loss of plasticity and various collapse phenomena in neural networks and suggest how symmetries can be used to design algorithms to enforce hard constraints in a differentiable way.

## 1 INTRODUCTION

Modern neural networks are so large that they contain an astronomical number of neurons and connections layered in a highly structured manner. This design of modern architectures and loss functions means that there are a lot of redundant parameters in the model and that the loss functions are often invariant to hidden, nonlinear, and nonperturbative transformations of the model parameters. We call these invariant transformations the "symmetries" of the loss function. Common examples of symmetries in the loss function include the permutation symmetry (Simsek et al., 2021; Entezari et al., 2021; Hou et al., 2019), rescaling symmetry (Dinh et al., 2017; Saxe et al., 2013; Neyshabur et al., 2014; Tibshirani, 2021), scale symmetry (Ioffe & Szegedy, 2015) and rotation symmetry (Ziyin et al., 2023b). In physics, symmetries are regarded as fundamental organizing principles of nature, and systems with symmetries exhibit rich and hierarchical behaviors (Anderson, 1972). However, a unifying theory is lacking to understand the role of symmetries in affecting the learning of neural networks and previous works often study specific symmetries case-by-case. Also, a predominant approach to analyzing symmetry often looks at symmetry with a negative light because symmetry is found to lead to saddle points, which slows down training (Li et al., 2019; Xiong et al., 2023). In this work, we take a neutral stance and show that the common types of symmetries can be understood in a unified framework through the lens of mirror symmetries, where every symmetry is proved to lead to a special structure and constraint of optimization.

Since we will also discuss stochastic aspects of learning, we study a generic twice-differentiable non-negative *per-sample loss function*:

$$\ell_\gamma = \ell_0(\theta, x) + \gamma\|\theta\|^2, \tag{1}$$

where $x$ is a minibatch or a single data point of arbitrary dimension and sampled from a training set. $\theta$ is the model parameter, and $\gamma$ is the weight decay. $\ell_0$ assumes the definition of the model architecture and is the data-dependent part of the loss. Training with stochastic gradient descent (SGD), we sample a set of $x$ and compute the gradient of the averaged per-sample loss over the set. The per-sample loss averaged over the training set is the empirical risk: $L_\gamma(\theta) := \mathbb{E}_x[\ell_\gamma]$. Training with gradient descent (GD), we compute the gradient with respect to $L_\gamma$. All the results we derive for $\ell_\gamma$ directly carry over to $L_\gamma$.

This work is organized as follows. We first study the effect of three specific types of symmetry one often encounters in deep learning: (1) rescaling symmetry, (2) rotation symmetry, and (3) permutation symmetry. See Figure 1 for an illustration. We then identify a general class of symmetry,

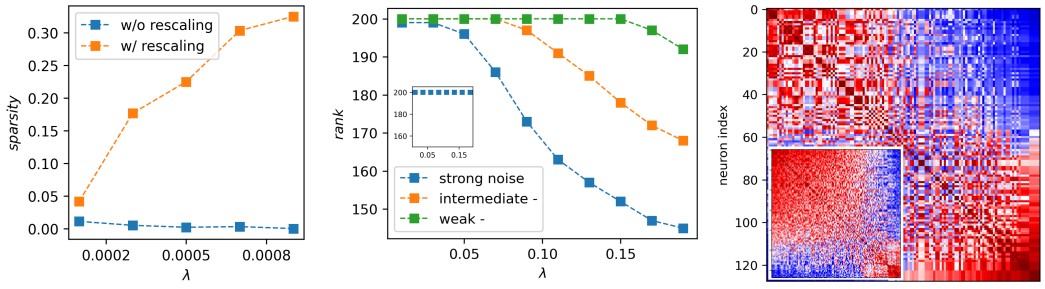

Figure 1: When loss function symmetries are present, the model converges to structurally constrained solutions at a high weight decay or gradient noise. **Left**: A vanilla linear regression trained with SGD does not converge to sparse solutions for any learning rate. When we introduce redundant rescaling symmetry to every parameter, sparser solutions are favored at higher learning rates ($\lambda$). **Mid**: Vanilla 200 dimensional matrix factorization trained with SGD prefers lower-rank solutions when the gradient noise is strong due to the rotation symmetry. The inset shows that the model always stays full-rank if we remove the rotation symmetry by introducing residual connections. **Right**: Correlation of the pre-activation value of neurons in the penultimate layer of ResNet18. After training, the neurons are grouped into homogeneous blocks when weight decay is present. The inset shows that such block structures are rare when there is no weight decay. Also, the patterns are similar for post-activation values, which further supports the claim that the block structures are due to the symmetry, not because of linearity. See Section 4.7 and A for the experimental details and more results.

*the mirror reflection symmetry*, that treats all three types of symmetry in a coherent framework and proves a general theorem showing that *every mirror symmetry leads to a structured constraint*, and when weight decay is used (or when the gradient noise is large) SGD training tends to converge to these constrained symmetric solutions. Here, a constraint refers to some function of the parameters being zero: $f(\theta) = 0$. Section 4 discusses the related works in detail and the connections of our results to them. All the proofs are given in Appendix B.

## 2 CONSEQUENCES OF COMMON SYMMETRIES

While all the theorems in this section can be proved as corollaries of the general theorem 4, we give independent proofs of them to bring some concreteness to the general theorem.

### 2.1 RESCALING SYMMETRY LEADS TO SPARSITY

The simplest type of symmetry in deep learning is the rescaling symmetry. Consider a loss function $\ell_0$ for which the following equality holds for any $x$, arbitrary vectors $u$, $w$ and $\rho \in \mathbb{R}_{/\{0\}}$:

$$\ell_0(u, w, x) = \ell_0(\rho u, \rho^{-1} w, x). \tag{2}$$

For the rescaling symmetry and for all the problems we discuss below, it is also possible for $\ell_0$ to contain other parameters $v$ that are irrelevant to the symmetry: $\ell_0 = \ell_0(u, w, v)$. Since having such $v$ or not does not change our result, we only show $v$ explicitly when necessary. Also, because the symmetries we consider in this work hold for any $x$, we also omit writing $x$ unless necessary.

The following theorem states that this symmetry leads to sparsity in the parameters.

**Theorem 1.** *Let $\ell_0(u, w)$ have the rescaling symmetry in Eq. (2). Then, for any $x$,*

1. *if $u = 0$ and $w = 0$, then $\nabla_u \ell_\gamma = 0$ and $\nabla_w \ell_\gamma = 0$;*
2. *for any fixed $u$, $w$, there exists $\gamma_0$ such that for all $\gamma > \gamma_0$, $\ell_\gamma(0,0) < \ell_\gamma(u,w)$.*

Two parts of the theorem statement convey different insights. Part 1 shows that the learning dynamics are constrained – namely, GD or SGD will not leave the condition $(u, w) = (0, 0)$ once entered. Part 2 shows that such constrained solutions can be locally favored for a large regularization. Additionally, symmetry has strong implications on the structures of the Hessian of the loss function and global properties of the loss landscapes; we delay its presentation and discussion after Theorem 4.

This symmetry usually manifests itself when part of the parameters is linearly connected. Previous works have used this property to either understand the inductive bias of neural networks or design efficient training algorithms. When the model is a fully connected ReLU network, Neyshabur et al. (2014) showed that having $L_2$ is equivalent to $L_1$ constraints of weights. Ziyin & Wang (2023) designed an algorithm to compress neural networks by transforming a parameter vector $v$ to $u \odot w$, where $\odot$ is the Hadamard product.

## 2.2 ROTATION SYMMETRY LEADS TO LOW-RANKNESS

A more involved but common type of symmetry is the rotation symmetry, which also appears in a few slightly different forms in deep learning. This type of symmetry appears in matrix factorization problems, where it is a main cause of the emergence of saddle points (Li et al., 2019). It also appears in Bayesian deep learning (Tipping & Bishop, 1999; Kingma & Welling, 2013; Lucas et al., 2019; Wang & Ziyin, 2022), self-supervised learning (Chen et al., 2020; Ziyin et al., 2023b), and transformers in the form of key-query matrices (Vaswani et al., 2017; Dong et al., 2021).

Now, we show that rotation symmetry in the landscape leads to low rankness. We use the word "rotation" in a broad sense, including all orthogonal transformations. There are two types of rotation symmetry common in deep learning. In the first kind, we have for any $W$,

$$\ell_0(W) = \ell_0(\Omega W) \tag{3}$$

for any orthogonal matrix $\Omega$ such that $\Omega \Omega^T = I$ and $W$ is a set of weights viewed as a matrix or vector whose left dimension matches the right dimension of $\Omega$.

**Theorem 2.** *Let $\ell_0$ satisfy the rotation symmetry in Eq. (3). Then, for any index $i$, vector $n$ and $x$,*

   1. *if $n^T W = 0$, then $n^T \nabla_W \ell_\gamma = 0$;*
   2. *for any fixed $W$, there exists $\gamma_0$ such that for all $\gamma > \gamma_0$, $\ell_\gamma(W_{/i}) < \ell_\gamma(W)$;*[1]

Part 1 of the statement deserves a closer look. $n^T W = 0$ implies that $W$ is low-rank and $n$ is a left eigenvector of $W$. That the gradient vanishes in this direction means that once the weight matrix becomes low-rank, it will always be low-rank for the rest of the training.

A more common symmetry is a "double" rotation symmetry, where $\ell_0$ depends on two matrices $U$ and $W$ and satisfies $\ell_0(U, W) = \ell_0(UR, R^T W)$, for any orthogonal matrix $R$ and any $U$ and $W$. Namely, the loss function is invariant if we simultaneously rotate two different matrices with the same rotation. In this case, one can show something similar: $n^T W = 0$ and $Un = 0$ for some fixed direction $n$ is the favored solution.

## 2.3 PERMUTATION SYMMETRY LEADS TO HOMOGENEITY

The most common type of symmetry in deep learning is permutation symmetry. It shows up in virtually all architectures in deep learning. A primary and well-studied example is that in a fully connected network, the training objective is invariant to any pairwise exchange of two neurons in the same hidden layer. We refer to this case as the "special permutation symmetry" because it is a special case of the permutation symmetry we study here. Many recent works are devoted to understanding the special permutation symmetry (Simsek et al., 2021; Entezari et al., 2021; Hou et al., 2019).

Here, we study a more general and abstract type of permutation symmetry. The loss function has a permutation symmetry between parameter subsets $\theta_a$ and $\theta_a$ if, for any $\theta_a$ and $\theta_b$,[2]

$$\ell_0(\theta_a, \theta_b) = \ell_0(\theta_b, \theta_a). \tag{4}$$

When there are multiple pairs that satisfy this symmetry, one can combine this pairwise symmetry to generate arbitrary permutations. In this perspective, permutation symmetries appear far more common than is recognized. For example, another example is that a convolutional neural network is invariant to a pairwise exchange of two filters, which is rarely studied. A scalar rescaling symmetry can also be regarded as a special case of permutation symmetry.

Here, we show that the permutation symmetry tends to make the neurons become identical copies of each other (namely, encouraging $\theta_a$ to be as close to $\theta_b$ as possible).

**Theorem 3.** *Let $\ell_0$ satisfy the permutation symmetry in Eq. (4). Then, for any $x$,*

   1. *if $\theta_a - \theta_b = 0$, then $\nabla_{\theta_a} \ell_\gamma = \nabla_{\theta_b} \ell_\gamma$;*
   2. *for any $\theta_a \neq \theta_b$, there exists $\gamma_0$ such that for all $\gamma > \gamma_0$, $\ell_\gamma((\theta_a + \theta_b)/2, (\theta_a + \theta_b)/2) < \ell_\gamma(\theta_b, \theta_a)$;*

---

[1]The notation $W_{/i}$ denotes the matrix obtained by setting the $i$-th singular value of $W$ to be zero.

[2]A common example is a hidden layer of a network; let $w_a$ and $u_a$ be the input and output weights of neuron $a$, and $w_b$, $u_b$ be the input and output weights of neuron $b$. We can thus let $\theta_a := (w_a, u_a)$ and $\theta_b := (w_b, u_b)$.

This theorem implies that a permutation symmetry can be seen as a generalized form of ensembling smaller submodels.[3] This identification of the stationary subspace agrees with the result in Simsek et al. (2021). Special cases of this result have been proved previously. For a fully connected network, Fukumizu & Amari (2000) showed that the solutions of subnetworks are also solutions of the larger network, and Chen et al. (2023) demonstrated that these subnetwork solutions of fully connected networks can be attractive when the learning rate is large. Our result is more general because it does not restrict to the special permutation symmetry induced by fully connected networks. A novel application is that the networks have block-wise neurons and activation patterns whenever weight decay is present. See Figure 1.

## 3 EVERY MIRROR SYMMETRY LEADS TO A STRUCTURED CONSTRAINT

A remarkable aspect of Theorems 1, 2 and 3 is that their proofs only require the symmetry, and no details of the architecture or loss function need to be specified. This means that these results are more general than the previous literature, which often specializes in a given architecture (such as a fully connected network) that happens to have a type of symmetry. The observation that only knowing the symmetry alone can help us deduce so much about the behavior of these systems hints at some underlying universal principle.

Let us first define a general type of symmetry called mirror reflection symmetry.

**Definition 1.** *A per-sample loss function $\ell_0(w)$ is said to have the simple mirror (reflection) symmetry with respect to a unit vector $n$ if, for all $w$, $\ell_0(w) = \ell_0((I - 2nn^T)w)$.*

Note that the vector $(I - 2nn^T)w$ is the reflection of $w$ with respect to the plane orthogonal to $n$. Also, the $L_2$ regularization term itself satisfies this symmetry for any $n$ because reflection is norm-preserving. An important quantity is the average of the two reflected solutions: $\bar{w} = (I - nn^T)w$, where $\bar{w}$ is the fixed point of this transformation and can be called a "symmetric solution." This mirror symmetry can be generalized to the case where the loss function is invariant only when multiple mirror reflections are made.

**Definition 2.** *Let $O$ consist of columns of orthonormal vectors: $O^T O = I$, and $R = I - 2OO^T$. A loss function $\ell_0(w)$ is said to have the O-mirror symmetry if, for all $w$, $\ell_0(w) = \ell_0(Rw)$.*

By construction, $OO^T$ and $I - OO^T$ are projection matrices, and $I - 2OO^T$ is an orthogonal matrix. There are a few equivalent ways to see this symmetry. First of all, it is equivalent to requiring the loss function to be invariant only after multiple simple mirror symmetry transformations. Let $m$ be a unit vector orthogonal to $n$. Reflections to both $m$ and $n$ give $(I - 2mm^T)(I - 2nn^T) = I - 2(nn^T + mm^T)$. The matrix $nn^T + mm^T$ is a projection matrix and, thus, an instantiation of $OO^T$. Secondly, because the composition of orthogonal unit vectors spans the space of projection matrices, $OO^T$ is nothing but a generic projection matrix $P$. Thus, this symmetry can be equivalently defined with respect to $P$ such that $\ell_0(w) = \ell_0((I - 2P)w)$. If we let $O$ or $P$ be rank-1, the symmetry reduces to the simple mirror symmetry in Definition 1.

We also make a reasonable smoothness assumption, which is only needed for part 4 of the theorem.[4]

**Assumption 1.** *The smallest eigenvalue of the Hessian of $\ell_0$ is lower-bounded by a (possibly negative) constant $\lambda_{\min}$.*

With these definitions, we are ready to prove the following theorem.

**Theorem 4.** *Let $\ell_0(w)$ satisfy the O-mirror symmetry. Then,*

1. *for any $\gamma$, if $O^T w = 0$, then $O^T \nabla_w \ell_\gamma = 0$;*
2. *if $O^T w = 0$, a subset of the eigenvector of $\nabla_w^2 \ell_0(w)$ spans $\ker(O^T)$, and the rest spans $\mathrm{im}(OO^T)$;*
3. *if $O^T w \neq 0$, there exists $\gamma_0$ such that for all $\gamma > \gamma_0$, $\ell_\gamma((I - OO^T)w) < \ell_\gamma(w)$;*

---

[3]One might suspect the origin is always favored when a mirror symmetry exists: this is not true. Let us consider a simple reparametrized linear regression problem: $L_\gamma(w_1, w_2) = [(w_1 + w_2)x - y]^2 + \gamma(w_1^2 + w_2^2)$. A permutation symmetry exists between $w_1$ and $w_2$. The condition $\theta_a - \theta_b = 0$ is satisfied for all solutions of the loss whenever $\gamma > 0$. Meanwhile, for a finite $\gamma$, no solution satisfies $\theta_a = \theta_b = 0$.

[4]Alternatively, we can assume that the parameters are constrained in a bounded space.

4. *there exists $\gamma_1$ such that for all $\gamma > \gamma_1$, all minima of $\ell_\gamma$ satisfy $O^T w = 0$.*

Parts 1 and 2 are statements regarding the local gradient geometry, regardless of the weight decay. Parts 3 and 4 are local and global statements regarding the role of weight decay. It is instructive to show how Theorems 1, 2 and 3 are corollaries of Theorem 4. The simplest application is to the rescaling symmetry. When the rescaling symmetry exists between two scalars $u$ and $w$, there are two planes of mirror symmetry: $n_1 = (1,1)$ and $n_2 = (1,-1)$. Here, $n_1$ symmetry implies that $u = -w$ is a symmetry solution, and $n_2$ symmetry implies that $u = w$ is a symmetry solution. Applying Theorem 4 to these two mirrors implies that $u = 0$ and $w = 0$ is a symmetry solution and obeys Theorem 1. When $u \in \mathbb{R}^{d_1}$ and $w \in \mathbb{R}^{d_2}$ are vectors of arbitrary dimensions and have the rescaling symmetry, one can identity the implied mirror symmetry as $O = I$, and so $I - 2P = -I$: the loss function is symmetric to a simultaneous flip of all the signs of $u$ and $w$. Applying Theorem 4 to this mirror again allows us to derive Theorem 1.

For permutation symmetry in $\ell_0(\theta_1, \theta_2)$ with $\theta_i \in \mathbb{R}^d$, we can identify the projection as

$$P = \frac{1}{2} \begin{bmatrix} I_d & -I_d \\ -I_d & I_d \end{bmatrix}. \tag{5}$$

Let $\theta = (\theta_1, \theta_2)$ denote a vector combination of both sets of the parameters. The permutation symmetry thus implies the mirror symmetry: $\ell_0(\theta) = \ell_0((I - 2P)\theta)$. The symmetry solution is $\theta_1 = \theta_2$, and applying the master theorem to this mirror allows us to obtain Theorem 3.

For rotation symmetry, we note that for any projection matrix $\Pi$, the matrix $I - 2\Pi$ is an orthogonal matrix because $(I - 2\Pi)(I - 2\Pi)^T = (I - 2\Pi)^2 = I$. Therefore, the rotation symmetry already implies that for any $\Pi$ and $W$, $\ell_0((I - 2\Pi)W) = \ell_0(W)$. To apply the theorem, we need to view $W$ as a vector, and the corresponding reflection matrix is $\mathrm{diag}(I - 2\Pi, ..., I - 2\Pi)$, a block-wise repetition of the matrix $I - 2\Pi$, where each block corresponds to a column of $W$. By construction, $P$ is also a projection matrix. Since this holds for an arbitrary $\Pi$, one can choose $\Pi$ to be the plane that matches the desired plane in Theorem 2, which can be then proved by invoking Theorem 4. Therefore, all three main types of symmetry we study are consequences of the general theorem.

## 4 APPLICATIONS

### 4.1 ABSORBING STATES AND STATIONARY CONDITIONS

To discuss the implication of symmetries, we introduce the concept of a "stationary condition."

**Definition 3.** *For an arbitrary function $f$, $f(\theta) = 0$ is a **stationary condition** of $L(\theta)$ if $f(\theta_t) = 0$ implies $f(\theta_{t+1}) = 0$, where $\theta_t$ is the $t$-th step parameter under (stochastic) gradient descent.*

A stationary condition can be seen as a special case of an absorbing state, which is a major theme in the study of Markov processes and is associated with complex phase-transition-like behaviors (Norris, 1998; Dickman & Vidigal, 2002; Hinrichsen, 2000). Part 1 of Theorem 4 implies the following.

**Corollary 1.** *Every $O$-mirror symmetry implies a linear stationary condition: $O^T \theta = 0$.*

Alternatively, a stationary condition can be seen as a generalization of a stationary point because every stationary point in the landscape implies the existence of a stationary condition – but not vice versa. For example, some functions of the parameters might reach stationarity before the whole model reaches stationarity. The existence of such conditions implies that there are special subspaces in the landscape such that the dynamics of gradient descent within these subspaces will not leave it. See Appendix Figure 4 for an illustration of the stationary conditions.

### 4.2 STRUCTURE OF THE HESSIAN

Part 2 of Theorem 4 hasd important implications for the local geometry of the loss and the dynamics of SGD. Let $H$ denote the Hessian of the loss $L$ or that of the per-sample loss $\ell$. Part 2 states that $H$ close to symmetry solutions are *partitioned* by the symmetry condition $I - 2P$ to two subspaces: one part aligns with the images of $P$, and the other part must be orthogonal to it. Namely, one can transform the Hessian into a two-block form, $H_\perp$ and $H_\parallel$, with $O$.[5] Note that the parameters

---

[5]Let $\tilde{O}$ be any orthogonal matrix whose basis includes all the eigenvectors of $O$. Then, $O^T H O$ will be a two-block matrix.

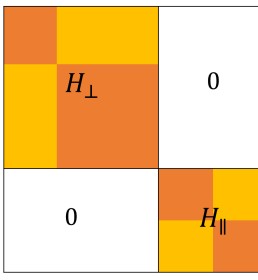 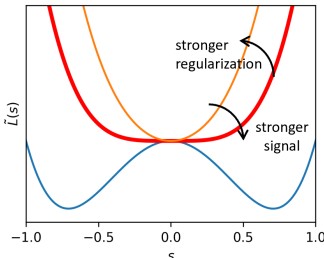 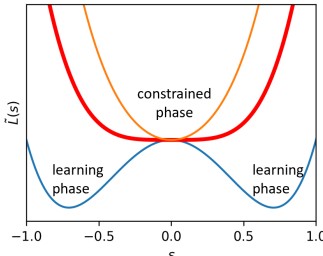

Figure 2: When symmetries exist, the stationary conditions correspond to highly structured Hessians. **Left**: the symmetry mirror $O$ partitions $H$ into two blocks: one block parallel to surfaces in $OO^T$, and the other orthogonal to it. When an extra symmetry exists, these two blocks can be decomposed into additional subblocks. **Mid-Right**: the loss function around a symmetric solution has a universal geometry. Here, $s$ is the component of the parameters along a direction of the $O$-symmetry. The competition between the signal in the dataset and the regularization strength determines the local landscape.

might also contain other symmetries, so $H_\parallel$ and $H_\perp$ may also consist of multiple sub-blocks. This implies that close to the symmetric solutions, the Hessian of the loss will take a highly structured form simultaneously for all data points or batches. See Figure 2.

That the Hessian of neural networks after training takes a similar structure is supported by empirical works. For example, the illustrative Hessian in Figure 2 is similar to that computed in (Sagun et al., 2016). That the actual Hessians after training are well approximated by smaller blocks is supported by (Wu et al., 2020). Blockwise Hessian matrices can also be related to the existence of gaps in the Hessian spectrum, which is widely observed (Sagun et al., 2017; Ghorbani et al., 2019; Wu et al., 2020; Papyan, 2018).

It is instructive to consider the special case where $O = n^T$ is rank-1. Part 2 implies that $n$ must be an eigenvector of the Hessian whenever the model is at a symmetry solution. For illustration, we consider a two-layer linear network with scalar input and outputs. The loss function can always be written as $\ell(w, u) = \frac{1}{2} \left( x \sum_i^d u_i w_i - y \right)^2$. For each index $i$, $u_i w_i$ contains the rescaling symmetry and are thus subject to two symmetries with mirrors $(1, 1)$ and $(1, -1)$. Therefore, the theory predicts that when $u \approx w \approx 0$, the Hessian consists of $d$ $2 \times 2$ symmetric matrices with $(1, 1)$ and $(1, -1)$ being the eigenvectors. This can be compared with a direct computation. When $w = u = 0$, the nonvanishing terms of the Hessian are $\frac{\partial^2}{\partial w_i \partial u_i} \ell = -xy$:

$$H = \begin{bmatrix} 0 & -xy & & & \\ -xy & 0 & & & \\ & & ... & & \\ & & & 0 & -xy \\ & & & -xy & 0 \end{bmatrix}. \tag{6}$$

This means that the eigenvectors are indeed $(1, 1)$ and $(1, -1)$ with eigenvalues $xy$ and $-xy$, agreeing with the theory. It is remarkable that we can identify all the eigenvectors by only examining the symmetry in the model.

### 4.3 DYNAMICS OF STOCHASTIC GRADIENT DESCENT

The symmetry in the loss has a lot of consequences for the dynamics of training with SGD in light of the recent progress in analyzing SGD. Let $O$ denote the mirror and $P = OO^T$ the projection matrix. If $OO^T w = sn$ where $n$ is a unit vector, and $s$ is a small quantity, the model is perturbatively away from the symmetry solution. In this case, one can expand the loss function to leading orders in $s$:

$$\ell(x, w) = \ell(x, w_0) + \frac{1}{2} w^T P H(x) P w + o(s^3), \tag{7}$$

where we have defined the sample Hessian restricted to the projected subspace: $H(x) := P \nabla_w^2 \ell(x, w_0) P$, which is a matrix of random variables. Note that all the odd-order terms in $s$ vanish due to the symmetry in flipping the sign of $s$. In fact, one can view the training loss $\ell_\gamma$ or $L_\gamma$ as a function of $s$, which we denote as $\tilde{L}(s)$, and this analysis implies that the loss landscape close to $s = 0$ takes a rather universal geometry. See Figure 2.

This allows us to characterize the dynamics of SGD in the symmetry directions:

$$P w_{t+1} = P w_t - \lambda H P w_t, \tag{8}$$

where $\lambda$ is the learning rate. Previously, this type of critical point is shown to exist at interpolation minima of wide networks (Wu et al., 2018). Our result implies that this type of solution is far more common than previously understood and exists whenever symmetries are present.

Let us first consider GD. The largest negative eigenvalue of $\mathbb{E}_x[H]$, $\xi^*$, thus gives the speed at which SGD escapes the stationary condition: $Pw_t \propto \exp(-\xi^* t)$. When weight decay is present, all the eigenvalues of $H$ will be positively shifted by $\gamma$, and, therefore, if and only if $\xi^* + \gamma > 0$, GD will be attracted to these symmetric solutions. In this sense, $\xi^*$ gives a critical weight decay value at which a symmetry-induced constraint is favored.

For SGD, the dynamics is qualitatively different. Naively, when using SGD, the model will escape the stationary condition faster due to the noise. However, this is the opposite of the truth. The existence of the SGD noise due to minibatch sampling makes these stationary conditions more attractive. The stability of the type of dynamics in Eq. (8) can be analyzed by studying the condition for convergence in probability of the solution $Pw = 0$ (Ziyin et al., 2023a). One can show that $Pw$ converges to $0$ in probability if and only if the Lyapunov exponent of the process $\Lambda$ is negative, which is possible even if this critical point is a strict saddle. When does a subspace of $Pw$ converge (or collapse) to zero? One can derive a satisfactory approximate learning rate by making the commutation approximation, which assumes that $H(x)$ commutes with $H(x')$ for all $x$, $x'$ in the training set.[6] In this case, each subspace of $H(x)$ has its own Lyapunov exponent and can be analytically computed. Let $\xi(x)$ denote the eigenvalue of $H(x)$ in this subspace. Then, this subspace collapses when $\Lambda = \mathbb{E}_x[\log|1 - \lambda(\xi(x) + \gamma)|] < 0$, which is negative for a large learning rate (see Appendix B for a formal treatment). The meaning of this condition becomes clear by expanding to the second order in $\lambda$ to obtain:

$$\lambda > \frac{-2\mathbb{E}[\xi + \gamma]}{\mathbb{E}[(\xi + \gamma)^2]}. \tag{9}$$

The numerator is the eigenvalue of the empirical loss, and the denominator can be identified as the minibatch noise effect (Wu et al., 2018), which becomes larger if the batch size is small or if the dataset is noisy. Therefore, this phenomenon happens due to the competition between the signal and noise in the gradient. This example shows that at a large learning rate, the stationary conditions are favored solutions of SGD, even if they are not favored by GD. From a Markovian perspective, this critical learning rate is when the Markov process becomes an absorbing Markov chain.[7] Also, convergence to these symmetry-induced saddles is not a unique feature of SGD but happens for Adam-type dynamics as well (Ziyin et al., 2021; 2023a).

Two novel applications of this analysis are to learning a sparse model and a low-rank model. See Figure 1. We first apply it to a linear regression with rescaling symmetry. It is known that when both weight decay and rescaling symmetries are present, the solutions are sparse and identical to lasso (Ziyin & Wang, 2023). Our result shows that even without weight decay, the solutions are sparse at a large learning rate. Then, we consider a matrix factorization problem. Classical results show that the solutions are low-rank when weight decay is present (Srebro et al., 2004). Our result shows that even if there is no weight decay, SGD at a large learning rate or gradient noise converges to these low-rank saddles. The fact that these constrained structures disappear completely when the symmetry is removed supports our claim that symmetry is the cause of them.

A strong piece of evidence for the relevance of the theory to real neural networks is that after training, the Hessian of the loss function is observed to contain many small negative eigenvalues, which hints at the convergence to saddle points (Sagun et al., 2016; 2017; Ghorbani et al., 2019; Alain et al., 2019). Another related phenomenon is that of pathological Fisher information. From a Bayesian perspective, the matrix $J := \mathbb{E}_x[\nabla_w \ell \nabla_w^T \ell]$ is the Fisher information of the system (Amari & Nagaoka, 2007). Our result implies that the Fisher information is singular close to any symmetry solutions. Note that $O^T \nabla_w \ell(w, x) = 0$ for a symmetry solution and any $x$. Therefore, the Fisher information has a zero eigenvalue along the directions orthogonal to any mirror symmetry. Previous works have demonstrated that the learning of neural networks passes through regions of singular

---

[6]We use this approximation to highlight its qualitative dependence on the learning rate, batch size, and gradient distribution. It should be noted that it is not an accurate approximation, and the computation and estimation of the Lyapunov exponent for this process is a well-known open problem for the field of dynamical systems (Pollicott, 2010).

[7]Alternatively, similar problems can also be analyzed using a continuous-time approximation and show that when gradient noise is strong, these points are attractive (Vivien et al., 2022; Chen et al., 2023).

Fisher information, where the learning dynamics is slow (Wei et al., 2008; Cousseau et al., 2008; Fukumizu, 1996; Karakida et al., 2019a;b). Therefore, the Fisher information having flat directions is also a sign that the symmetry solutions are reached.

### 4.4 LOSS OF PLASTICITY AND NEURAL COLLAPSES

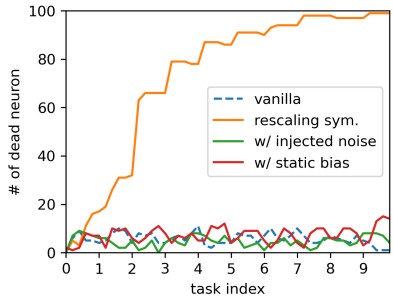

Our theory implies that the commonly observed loss of plasticity problem in continual and reinforcement learning (Lyle et al., 2023; Abbas et al., 2023; Dohare et al., 2023) is attributable to symmetries in the model. For a given task, weight decay or a finite learning rate makes the model converge to symmetry solutions, which tend to be low-capacity constrained solutions. If we train on an additional task, the capacity of the model can only decrease because the symmetry solutions are also stationary conditions, which SGD cannot escape. Fortunately, our theory suggests at least two ways to fix this problem: (1) use an alternative parameterization that explicitly removes the symmetry and/or (2) inject additive noise to the gradient to eliminate the stationary conditions.[8] There are many ways to achieve (1). An easy way is to bias every (symmetry-relevant) parameter by a random bias: $w_i \to w_i + \beta_i$, where $\beta_i$ is a small fixed random variable.[9] See Figure 3 and Appendix A.

Figure 3: Loss of plasticity in continual learning in a vanilla linear regressor (**dashed**) and linear regressors with rescaling symmetry (**solid**). Vanilla regression has no symmetry and does not suffer plasticity loss, whereas having symmetries leads to the loss of plasticity. One can fix the problem with one of the two suggested methods, either by removing the symmetry in the model or removing the absorbing states by injecting noise.

A related phenomenon that symmetry can explain is the collapse of neural networks. The most common type of collapse is when the learned representation of a neural network spans a low-rank subspace of the entire available space, often leading to reduced expressive power. In Bayesian deep learning, a posterior collapse happens when the stochastic latent variables are low-rank (Lucas et al., 2019; Wang & Ziyin, 2022). This can be attributed to the double rotation symmetry of the encoder's last layer weight and the decoder's first layer weight. In self-supervised learning, a dimensional collapse happens when the representation of the last layer is low-rank (Tian, 2022), which has been found to be explained by the rotation symmetry of the last layer weight matrix. This also explains why many self-supervised learning methods focus on removing the symmetry (Bardes et al., 2021). The rank collapse that happens in self-attention may also be relevant (Dong et al., 2021). In supervised learning, the "neural collapse" happens when the learned representation of the penultimate learning becomes low-rank, which happens when weight decay is present (Papyan et al., 2020). Figure 1 shows that such a phenomenon can be attributed to the permutation symmetry in the fully connected layer. In summary, our result provides a unified perspective of the collapse phenomenon: collapses are caused by symmetries in the loss function. Our theory also suggests that these collapse phenomena have a natural interpretation as "phase transitions" in theoretical physics, where a collapse solution corresponds to a symmetric state.

### 4.5 $L_1$ EQUIVALENCE OF MIRROR SYMMETRIES

Parts 3 and 4 of Theorem 4 imply that constrained solutions are favored when weight decay is used. These results can be stated in an alternative way: that *every mirror symmetry plus weight decay has an $L_1$ equivalent*. To see this, let the loss function $L_0(w)$ be $O$-symmetric, and $P = OO^T$. Let $w$ be an arbitrary weight, which we decompose as $w = w' + sPw/\|Pw\|$, where we define $s = \|Pw\|$. Let us define an equivalent loss function $\tilde{L}_0(w', Pw/\|Pw\|, s^2) := L_0(w)$. By definition, we have

---

[8]In fact, gradient noise injection is a known method to alleviate plasticity loss (Dohare et al., 2023).

[9]The effectiveness of this method raises an interesting question of why it works because adding biases only removes mirror symmetries that pass through the origin, and symmetries with respect to a hyperplane still exist. In short, it is usually not symmetries that lead to bad solutions but that these symmetries appear together with small-norm solutions that are low-capacity. This is true for $O$-mirror symmetries, which pass through the origin. However, when the symmetry does not intersect the origin, the symmetric solutions do not coincide with small-norm solutions (which are preferred when there is weight decay), and are thus benign. For example, consider the loss $\ell = (uwx - y)^2$. Here, the symmetry solution is $u = w = 0$, which is also low-capacity. When we add a bias, say $\ell = ((u - 0.1)wx - y)^2$, the symmetric solution no longer coincides with a small-norm solution, and adding weight decay actually prevents collapsing to the symmetric solution.

successfully constructed the $L_1$ equivalent of the original loss.

$$L_0(w) + \gamma\|w\|^2 = \tilde{L}_0(w', Pw/\|Pw\|, s^2) + \gamma(\|w'\|^2 + s^2) = \tilde{L}_0(w', Pw/\|Pw\|, |z|) + \gamma(\|w'\|^2 + |z|),$$

where we introduced $|z| = s^2$. Therefore, along the symmetry-breaking direction, the loss function has an equivalent $L_1$ form. One can also show that $\tilde{L}_0$ is well defined as an $L_1$-constrained loss function. If $L_0$ is differentiable, $\tilde{L}_0$ is differentiable except at $s = 0$. Thus, it suffices to show that the right derivative of $\tilde{L}_0$ with respect to $z$ exists at $z = 0_+$. As we have discussed, at $z = 0$, the expansion of $L_0$ is second order in $s$. This means that the leading order term of $\tilde{L}_0$ is first order in $z$, and so the $L_1$ penalty is well-defined for this loss function.

## 4.6 An Algorithm for Differentiable Constraint

Sparsity and low-rankness are typical structured constraints that practitioners often want to incorporate into their models (Tibshirani, 1996; Meier et al., 2008; Jaderberg et al., 2014). However, the known methods of achieving these structured constraints tend to be tailored for specific problems and based on nondifferentiable operations. Our theory shows that incorporating symmetries is a general and scalable way to introduce such constraints into deep learning. Consider solving the following constrained problem: $\min_\theta L(\theta)$ *s.t. as many elements of $P\theta$ are zero as possible*. Here, $P = OO^T$ is a projection matrix. Our theory implies an algorithm for enforcing such constraints in a differentiable way: introducing an artificial $O$-symmetry to the loss function encourages the constraint $O^T\theta = 0$, which can be achieved by running GD on the following loss function:

$$\min_{w,u,v} L(T(w, u, v)) + \alpha(\|w\|^2 + \|u\|^2), \tag{10}$$

where $w$, $u$, $v$ have the same dimension as $\theta$ and $T(w, u, v) = (I - P)v + (Pw) \odot (Pu)$, where $\odot$ denotes the Hadamard product. We call the algorithm *DCS*, standing for differentiable constraint by symmetry. This parameterization introduces the mirror symmetry to which $O^T T(w, u, v) = 0$ is a stationary condition. By Theorem 4, a sufficiently large $\alpha$ ensures that $O^T T(w, u, v) = 0$ is an energetically favored solution. Also, note that this parametrization is a "faithful" parametrization in the sense that it is always true that $\min_{w,u,v} L(T(w, u, v)) = \min_\theta L(\theta)$. See Section A for an application of the algorithm to ResNet18.

## 4.7 Numerical Results

We numerically illustrate the effects our theory implies. All the technical details of the experiments are presented in Section A. We first conduct experiments relating to the rescaling symmetry. See Figure 1-left and 3. Here, we consider a linear regression task with noisy Gaussian data, where the loss function is $\ell = (v^T x - y)$, where $v$ is either directly trained or parametrized as the Hadamard product of two parameter vectors to artificially introduce rescaling symmetry: $v = u \odot w$. We see that without such symmetry, the model never converges to a sparse solution, whereas the symmetrized parametrization converges to symmetry solutions. Figure 1-mid shows that low-rank solutions are preferred in matrix factorization when the gradient noise is large, whereas such a tendency disappears when one removes the rotation symmetry by introducing a residual connection. Figure 1-right shows that homogeneous solutions are preferred when weight decay is used, in agreement with the prediction of Theorem 4.

## 5 Discussion

In this work, we studied the implications of loss function symmetries on the gradient-based learning of models. We have shown that every mirror symmetry leads to a structured constraint of learning. This statement is examined from two different angles: (1) such solutions are favored when $L_2$ regularizations are applied; (2) they are favored when the gradient noise is strong (which can happen when the learning rate is large, the batch size is small, or the data is noisy). We showed that the theory can analyze and understand common structures such as sparsity and low-rankness. We also discussed a variety of specific problems and phenomena in a unified manner. Our result is universal in that it only relies on the existence of the specified symmetries and does not rely on the properties of the loss function, model architectures, or data distributions. Per se, symmetry and its associated constraint are both good and bad. On the bad side, it limits the expressivity of the network and its approximation power. On the good side, it leads to more condensed models and representations, tends to ignore features that are noisy and can improve generalization capability thereby. Understanding symmetry systematically can help us avoid its negative side and utilize it to our advantage.

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

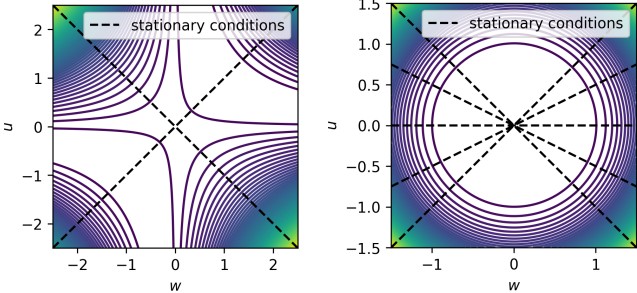

Figure 4: Stationary conditions in different loss landscapes. **Left**: $L = (wu - 1)^2$. Here, $u = w$ and $u = -w$ are the stationary conditions caused by the rescaling symmetry. **Right**: $\theta = (u, w)$ and $L = -||\theta||^2 + ||\theta||^4$. Here, the stationary condition caused by the rotation symmetry is every straight line crossing the origin. Every stationary condition delineates a submanifold of the entire landscape. Once the model is in this submanifold, SGD cannot leave it.

## A  EXPERIMENTAL CONCERNS

### A.1  ILLUSTRATION OF STATIONARY CONDITIONS

See Figure 4.

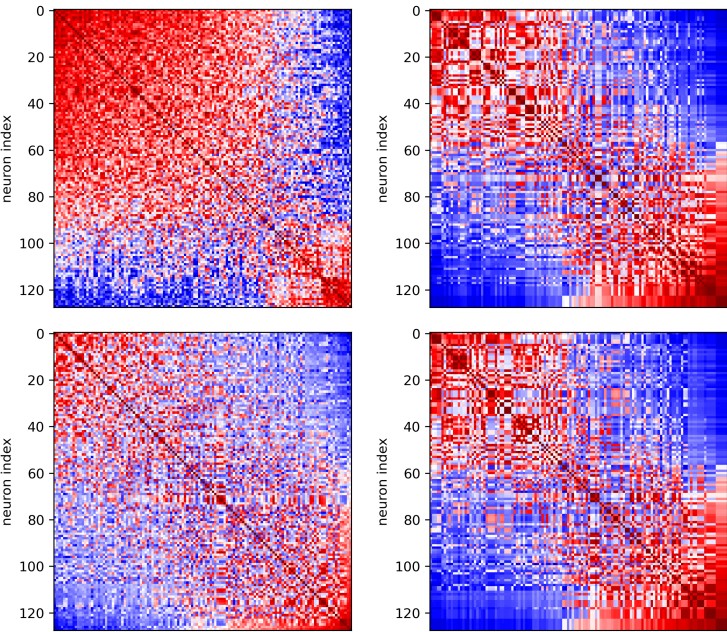

Figure 5: Comparison for the correlation matrix of the neurons in the penultimate layer at zero weight decay (**left**) and 0.001 weight decay (**right**). **Upper**: pre-activation correlation. **Lower**: post-activation correlation. After training, the neurons are grouped into homogeneous blocks when weight decay is present. The inset shows that such block structures are very rare when there is no weight decay. Also, the patterns are similar for post-activation values, which further supports the claim that the block structures are due to the symmetry, not because of linearity.

## A.2 EXPERIMENTAL DETAILS AND ADDITIONAL RESULTS FOR FIGURE 1

Here, we give the experimental details for the experiments in Figure 1.

For the sparsity experiments, we generate online data of batch size 1 in the following way. The input $x \in \mathbb{R}^{200}$ is sampled from a diagonal normal distribution. The label $y = \frac{1}{200} \sum_i x_i + \epsilon \in \mathbb{R}$, where $\epsilon$ is a noise term also sampled from a Gaussian distribution. The training proceeds with SGD without weight decay or momentum for $10^5$ iterations. The vanilla linear regression (labeled as "w/o rescaling") is parameterized in the standard way: $f(x) = w^T x$. The regressor with rescaling symmetry is parameterized as a Hadamard product, as in the spred algorithm (Ziyin & Wang, 2023): $f(x) = (w \odot u)^T x$, where $\odot$ denotes the element-wise product.

For the low-rank experiment with matrix factorization, we also generate online data of batch size 1 similarly. The input $x \in \mathbb{R}^{200}$ is sampled from a diagonal normal distribution. The label is $y = \mu x + (1-\mu)\epsilon \in \mathbb{R}^{200}$, where $\mu$ controls the degree of noise in the label and can be seen as the effective signal-to-noise ratio in the data. Here, the noise vector $\epsilon$ have different variances: $\epsilon_i \sim \mathcal{N}(0, 2/i)$. The vanilla matrix factorization model is $f(x) = WUx$, where both $W$ and $U \in \mathbb{R}^{200 \times 200}$. The training proceeds with standard SGD without momentum or weight decay. For the inset figure, we parameterize the network through residual connections: $f(x) = (I_{200} + W)(I_{200} + U)x$, thus removing the rotation symmetry.

For the ResNet experiment, we train a standard ResNet18 with roughly 10M parameters in total on the CIFAR-10 dataset. The SGD algorithm uses a batch size of 128 for 100 epochs with a fixed learning rate of 0.1 and momentum of 0.9, with varying degrees of weight decay. To plot the activation correlation, we take the penultimate layer neurons of the fully connected layer with dimension 128 and compute the correlation matrix over their activation of 2000 unseen test points. The neurons are sorted according to the eigenvector with the largest eigenvalue of the correlation matrix to reveal its block structure. Importantly, the pre- and post-activations have a similar correlation structure, showing that the effect is not due to linearity but the permutation symmetry. See Figure 5 for the comparison between the pre- and post-activation correlations.

### A.3 Experimental Detail for Continual Learning

Here, we give the experimental detail for the continual learning experiment in Figure 3.

For all the experiments in the figure, the training proceeds with Adam without momentum with a batch size of 16 for 25000 steps. Every task consists of a dataset of 100 data points drawn from the following distribution. The input $x \in \mathbb{R}^{100}$ is sampled from a diagonal normal distribution. The label $y = \frac{1}{100} \sum_i x_i + \epsilon \in \mathbb{R}$, where $\epsilon$ is a noise term also sampled from a Gaussian distribution. The weights obtained from training on task $j$ is used as the initialization for task $j + 1$, which consists of another 100 data points sampled in the same way. We train for 10 tasks and record the number of dead neurons in the model. The dead neurons are defined as the number of parameters that have a vanishing gradient.

To have strong control over the experimental conditions, we use vanilla linear regression as a base model, which is shown in the solid curve. Because there is no symmetry in the model, the vanilla linear regression has a minimal level of dead neurons, and its number does not increase as the number of tasks increases.

In contrast, for a linear regression with augmented rescaling symmetry where we reparameterize every weight of the linear regressor by the Hadamard product of two independent weights (also see the previous section), the loss of plasticity problem emerges, and the number of dead neurons increases steadily as one train on more and more tasks. To show that symmetry is indeed the cause of the problem, we fix the loss of plasticity problem in this model with the two suggested methods. First, we inject a very weak Gaussian random noise with variance $1e - 4$ to the gradient every step. Because this removes the absorbing states, or equivalently the stationary conditions, the number of dead neurons reduces to the same level as vanilla regression. Alternatively, we bias every weight parameter by a random and fixed constant: $w_i \rightarrow w_t + \beta_i$, where $\beta_i$ is drawn from a Gaussian distribution with variance $1e - 4$. Because this parametrization removes the symmetry in the model, it also fixes the loss of plasticity problem, as we expect from the theory.

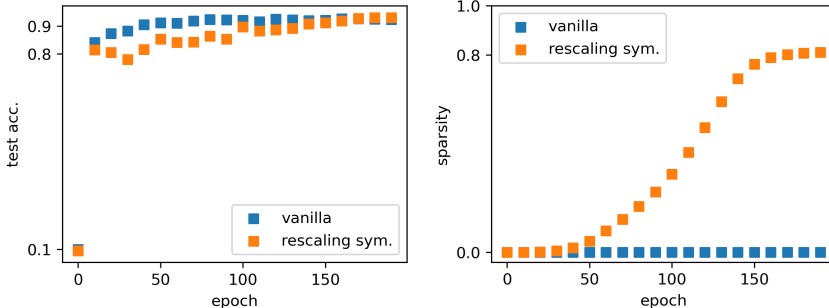

Figure 6: Training of a ResNet18 on CIFAR-10 without (vanilla) and with rescaling symmetry on each parameter. **Left**: the two models are similar in terms of training time and final performance. **Right**: with rescaling symmetry, the model parameters is very sparse. Here, sparsity is defined as the fraction of parameters with a magnitude smaller than $10^{-6}$. Setting these parameters to zero has no discernible effect on the model performance.

## A.4 LEARNING DYNAMICS OF THE DCS ALGORITHM

To demonstrate the learning dynamics of the DCS algorithm, We consider training a sparse ResNet18 on CIFAR-10. Here, the training proceeds with SGD with $0.9$ momentum and batch size $128$, consistent with standard practice. We use a cosine learning rate scheduler for $200$ epochs. We compare the learning dynamics of vanilla ResNet18 and a ResNet18 with the rescaling symmetry on every parameter, where we reparametrize the original parameter vector $v$ as the Hadamard product of two vectors $w \otimes v$. Both models use a weight decay of 5e-4. We note that this special case of the DCS algorithm is identical to the spred algorithm (Ziyin & Wang, 2023). After training, both the vanilla model and the DCS model reach roughly $93\%$ test accuracy (with the DCS model higher by a small margin).

See Figure 6. As is clear, the training time required for a DCS model is similar to that of a vanilla model. In terms of memory cost, we note that DCS costs twice as much memory as the vanilla at batch size 1. However, at the batch size 128, the memory cost difference between the two is smaller than 10 percent.

## B  Theoretical Concerns

### B.1  A Formal Derivation of Eq. (9)

By Definition 2, the loss function has the $O$-symmetry if for any $x$

$$\ell(w, x) = \ell(I - 2OO^T w, x). \tag{11}$$

As we discussed, this means that for every data point $x$, the per-sample Hessian $\nabla_w^2 \ell(w, x)$ takes the same block-wise structure outlined in Fig. 2. For this chapter, the most important consequence of Theorem 4 is that $O^T w = 0$ is a symmetry solution of $\ell(w, x)$ for all $x$.

We are interested in the stability of these solutions. Namely, we are interested in whether the model will be attracted back to the solution if we are perturbatively away from it. The expansion of the per-sample loss to the second order gives:

$$\ell(w, x) = \ell(w^{(0)}, x) + \frac{1}{2} w^T P \nabla_w^2 \ell(w^{(0)}, x) P w + o(s^4), \tag{12}$$

where $P = OO^T$ is a projection matrix, and $w^{(0)} = Pw$ is the component of $w$ that is orthogonal to the symmetry breaking subspace. Here, we care about when $Pw$ is attracted towards $0$. The dynamics of $z := Pw$ is thus a stochastic linear dynamics:

$$z_{t+1} = z_t - \lambda \hat{H}(w^{(0)}, x) z_t, \tag{13}$$

where $\hat{H}(w^{(0)}, x) = P \nabla_w^2 \ell(w^{(0)}, x) P$.

To proceed, we make the following assumption.

**Assumption 2.** *(Stationary background dynamics) The motion of $w_0$ is sufficiently slow that $\hat{H}(w^{(0)}, x) = \hat{H}^*(x)$ is a constant function in $w^{(0)}$.*

This also implies that any eigenvalue of $H$ also only depends on $x$. This assumption holds when the time scale of relaxation for $w^{(0)}$ is far slower than that of $Pw$ or when the dynamics is already stationary, namely, close to convergence.

When Assumption 2 holds, and $O$ is rank-1, this dynamics is analytically solvable. By Theorem 4, if $O = n$ is rank-1, $n$ is an eigenvector of $\hat{H}$ for all $x$. Thus, the dynamics simplifies to a one-dimensional dynamics, where $h(x) \in \mathbb{R}$ is the corresponding eigenvalue of $\hat{H}(w_0, x)$:

$$z_{t+1} = z_t - \lambda h(x) z_t. \tag{14}$$

The sufficient and necessary condition for the stability of this dynamics at $z = 0$ has an analytical solution (Ziyin et al., 2023a), which is Eq. (9).

**Theorem 5.** *(Ziyin et al. (2023a)) Let $w_t$ follow Eq. (14). Then, for any data set,*

$$w_t \to_p 0 \tag{15}$$

*if and only if[10]*

$$\mathbb{E}_x[\log|1 - \lambda h(x)|] < 0. \tag{16}$$

### B.2  Proofs

#### B.2.1  Proof of Theorem 1

*Proof.* We first show part 1. The rescaling symmetry states that for any $\epsilon \neq 1$ and $w$, $u$,

$$\ell_0((1 + \epsilon)u, w/(1 + \epsilon)) = \ell_0(u, w). \tag{17}$$

For an infinitesimal $\epsilon$, this condition leads to

$$\nabla_w \ell_0 \cdot w = \nabla_u \ell_0 \cdot u. \tag{18}$$

---

[10]This condition generalizes to the case when the batch size $S$ is larger than 1, where $h(x)$ becomes the per-batch Hessian, and the expectation is taken over all possible batches.

Taking the derivative of both sides over $w$, we obtain

$$\nabla_w \ell_0 = -\nabla_w^2 \ell_0 \cdot w + \nabla_w \nabla_u \ell_0 \cdot u. \tag{19}$$

Therefore, the gradient of $\ell_\gamma$ is $-\nabla_w^2 \ell_0 \cdot w + 2\gamma w + \nabla_w \nabla_u \ell_0 \cdot u$. When both $w$ and $u$ are zero, $\nabla_w \ell_\gamma = 0$. Likewise, we can show that $\nabla_u \ell_\gamma = 0$. This proves part 1.

For part 2, let us denote the quantity $\ell_\gamma(0,0) - \ell_\gamma(u,w)$ as $\Delta$. Now, note that $\Delta = \ell_0(0,0) - \ell_0(u,w) - \gamma(\|u\|^2 + \|w\|^2)$, and so setting

$$\gamma > \max\left(0, \frac{\ell_0(0,0) - \ell_0(u,w)}{\|u\|^2 + \|w\|^2}\right) \tag{20}$$

fulfills the requirement. Note that because $\ell_0$ is differentiable, the fraction always exists. This proves part 2.

$\square$

### B.2.2 Proof of Theorem 2

*Proof.* We focus on proving part 1. For an arbitrary and fixed index, $i$, of the singular values of $W$, we consider a continuous transformation of $W_0 = W(s)$. Define a diagonal matrix $\tilde{\Sigma}_{jj} = \Sigma_{jj}$ for all $j \neq i$, and define

$$\tilde{\Sigma}_{jj}(s) = \begin{cases} \Sigma_{jj} & \text{if } j \neq i; \\ s\Sigma_{jj} & \text{if } j = i. \end{cases} \tag{21}$$

We also define a transformed version of $V$, which depends on an arbitrary vector $z$:

$$\tilde{V}_{kl}(z) = \begin{cases} V_{kl} & \text{if } k \neq i; \\ z_l & \text{if } k = i. \end{cases} \tag{22}$$

With $\tilde{\Sigma}$ and $\tilde{V}$, we define $\tilde{W}$

$$\tilde{W}(s,z) = U\tilde{\Sigma}(s)\tilde{V}. \tag{23}$$

We note two different features of this transformation: (1) $W(0)$ is low-rank, and (2) for any $s$, $\ell(W(s)) = \ell(W(-s))$. To see this, note that there exists an orthogonal matrix $R$ such that

$$RW(s) = W(-s). \tag{24}$$

By the assumed symmetry of the loss function, we have $\ell(W(s)) = \ell(RW(s)) = \ell(W(-s))$. Because

$$\frac{d}{ds}W_{jk}(s,z) = U_{ji}\Sigma_{ii}\tilde{V}_{ik}(z) = U_{ji}\Sigma_{ii}z_k, \tag{25}$$

we can take the derivative of $s$ of both sides of the equality $\ell(W(s)) = \ell(W(-s))$ to obtain a low-rank condition on the gradient width as a matrix:

$$\Sigma_{ii}\sum_{jk}\left[\nabla_{W_{jk}}\ell(W(s)) + \nabla_{W_{jk}}\ell(W(-s))\right]U_{ji}z_k = 0. \tag{26}$$

In the limit $s \to 0$, $W(s) = W(-s)$ and so the equality leads to

$$2\Sigma_{ii}\sum_{jk}\nabla_{W_{jk}}L(W(0))U_{ji}z_k = 0. \tag{27}$$

Because this equality must hold for any $z_k$, we have that $U_{ji}$ must be a left eigenvector of $\nabla_{W_{jk}}\ell(W(0))$ with zero eigenvalues. Substituting into the gradient descent algorithm, we have

$$\sum_j U_{ji}W_{jk,t+1} = \sum_j U_{ji}W_{jk,t} - \lambda \sum_j U_{ji}\nabla_{W_{jk}}\ell(W_t) = 0. \tag{28}$$

This proves part 1.

For part 2, we note that the Frobenious norm of a matrix is the sum of its squared singular values. Therefore, if we hold other singular values unchanged and shrink one of the singular values to $0$, the $L_2$ regularization part of the loss function will strictly decrease. The rest of part 2 is the same as the proof of Theorem 1. $\square$

### B.2.3   PROOF OF THEOREM 3

*Proof.* The symmetry condition is

$$\ell_0(\theta_a, \theta_b) = \ell_0(\theta_b, \theta_a). \tag{29}$$

Taking the gradient of both sides with respect to $\theta_a$, we obtain

$$\nabla_{\theta_a}\ell_0(\theta_a, \theta_b) = \nabla_{\theta_a}\ell_0(\theta_b, \theta_a). \tag{30}$$

When $\theta_a = \theta_b$, we can write the above condition as

$$\nabla_{\theta_a}\ell_0(\theta_a, \theta_b) = \nabla_{\theta_b}\ell_0(\theta_a, \theta_b). \tag{31}$$

This proves the first part of the theorem.

We now prove the second part of the theorem. Let us define interpolation functions $g_a$ and $g_b$:

$$g_a(\mu) = (0.5 - \mu)\theta_a + (0.5 + \mu)\theta_b; \tag{32}$$

$$g_b(\mu) = (0.5 + \mu)\theta_a + (0.5 - \mu)\theta_b. \tag{33}$$

With these definitions, we have $g_a(\mu) = g_b(-\mu)$. Also, we note that

$$g_a(0) = g_b(0) = 0.5\theta_a + 0.5\theta_b, \tag{34}$$

which is the solution we want to compare with.

The loss function is given by

$$\ell_\gamma(\theta_a, \theta_b) = \ell_\gamma(g_a(0.5), g_b(0.5)). \tag{35}$$

In contrast, for the homogeneous solution, the loss value is

$$\ell_\gamma(g_a(0), g_b(0)). \tag{36}$$

The norms of the two solutions, $\mu = 0.5$ and $\mu = 0$, can be compared:

$$\Delta := \|g_a(0)\|^2 + \|g_b(0)\|^2 - \|g_a(0.5)\|^2 + \|g_b(0.5)\|^2 < 0, \tag{37}$$

where the inequality follows from the Cauchy-Schwarz inequality and the assumption that $\theta_a \neq \theta_b$. Therefore, for any

$$\gamma > \frac{\ell_0(g_a(0.5), g_b(0.5)) - \ell_0(g_a(0), g_b(0))}{\Delta}, \tag{38}$$

$\ell_\gamma(g_a(0), g_b(0)) < \ell_\gamma(\theta_a, \theta_b)$. This proves the second part of the statement. $\qquad \square$

### B.2.4   PROOF OF THEOREM 4

*Proof.* Part 1. Let $R := (I - 2OO^T)$. By assumption, we have $O^T w = 0$. Now, consider a linearly transformed version of $w$:

$$\tilde{w}(s) = w + sn, \tag{39}$$

where $n$ is any unit vector in the image of $OO^T$. Note that we have the following relation:

$$R\tilde{w}(s) = (I - 2OO^T)(w + sn) = w - sn = \tilde{w}(-s). \tag{40}$$

Therefore, by definition of the mirror symmetry, we have that for all $s$:

$$\ell_\gamma(\tilde{w}(s)) = \ell_\gamma(\tilde{w}(-s)). \tag{41}$$

Dividing both sides by $s$ and taking the limit $s \to 0$, we obtain

$$n^T \nabla_w \ell_\gamma(w) = 0. \tag{42}$$

Because $n$ is arbitrary, one can select a set of $n$ such that they span the rows of $O^T$, and we obtain that $O^T \nabla_w \ell_\gamma(w) = 0$. This finishes part 1.

Part 2. Let $O^T w = 0$. By symmetry, we have that for any $s \in \mathbb{R}$ and $n \in \ker(O^T)^{\perp}$:[11]

$$\ell_0(w + sn) = \ell_0(w - sn). \tag{43}$$

---

[11]We use $\ker(O^T)^{\perp}$ to denote the set of all vectors that is perpendicular to all the vectors in $\ker(O^T)$.

Let $m$ be an arbitrary vector in $\ker(O^T)$. Then, we also have that for any $s' \in \mathbb{R}$

$$\ell_0(w + sn + s'm) = \ell_0(w - sn + s'm). \tag{44}$$

Taking derivative over $s'$ for both sides and let $s' \to 0$, we obtain

$$m^T \nabla \ell_0(w + sn) = m^T \nabla \ell_0(w - sn). \tag{45}$$

Taking derivative over $s$ and let $s \to 0$, we obtain

$$2m^T \nabla_w^2 \ell_0(w) n = 0. \tag{46}$$

Since $m$ is an arbitrary vector in $\ker(O^T)$ and $n$ is an arbitrary in $\ker(O^T)^\perp$, this implies that

$$\nabla_w^2 \ell_0(w) n \in \ker(O^T)^\perp, \tag{47}$$

$$\nabla_w^2 \ell_0(w) m \in \ker(O^T). \tag{48}$$

Namely, a subset of the eigenvectors of $\nabla_w^2 \ell_0(w)$ spans $\ker(O^T)^\perp$ and the rest spans $\ker(O^T)$. This proves part 2.

To prove part 3, we first recognize that if we only look at the $L_2$ regularization part of the loss function, an orthogonal solution is always favored over a non-orthogonal solution. Let $w$ be an arbitrary solution such that $O^T w \neq 0$. We decompose $w$ into an orthogonal part and a non-orthogonal part:

$$w = u + sn, \tag{49}$$

where $O^T u = 0$ and $OO^T n = n$. Since $u$ and $n$ are orthogonal, we have that

$$\|w\|^2 - \|u\|^2 = s^2 > 0. \tag{50}$$

Therefore, if

$$\gamma > \frac{\ell_0(u) - \ell_0(w)}{s^2}, \tag{51}$$

we have that

$$\ell_\gamma(w) - \ell_\gamma(u) = \ell_0(w) - \ell_0(u) + \gamma(\|w\|^2 - \|u\|^2) \tag{52}$$

$$= \ell_0(w) - \ell_0(u) + \gamma s^2 \tag{53}$$

$$> \ell_0(w) - \ell_0(u) + \frac{\ell_0(u) - \ell_0(w)}{s^2} s^2 = 0. \tag{54}$$

However, since we have $u = (I - OO^T)w$, this proves part 3.

Part 4. By assumption, the smallest Hessian eigenvalue of $\ell_0$ is lower bounded by $\lambda_{\min}$. Therefore, if $\gamma > \lambda_{\min}$, $\ell_\gamma$ has a positive definite Hessian everywhere, implying that its gradients are monotone and that the global minimum is unique. Now, suppose there exists $u = w + c_0 n$ such that $c_0 \neq 0$, $O^T w = 0$, $OO^T n = n$, and

$$\nabla \ell_\gamma(u) = 0. \tag{55}$$

Then,

$$n^T \nabla \ell_\gamma(u) = 0 = n^T \nabla \ell_\gamma(w). \tag{56}$$

This implies that the gradient is not monotone, which contradicts the assumption. Therefore, we have proved part 4. $\qquad \square$

