# OpenReview forum: "Symmetry Leads to Structured Constraint of Learning"
_ICLR.cc/2024/Conference — Submitted to ICLR 2024_

### Official Review · Reviewer_Wp9T · 2023-10-20

**Soundness:** 2 fair
**Presentation:** 3 good
**Contribution:** 3 good
**Rating:** 5
**Confidence:** 3

**Summary:**

This paper studies how loss function symmetries affect learning. The authors propose a general definition, namely mirror reflection symmetry, which unifies common symmetries including rescaling, rotation, and permutation symmetry. They then prove that every mirror symmetry leads to a structured constraint, which becomes a favored solution when the weight decay or gradient noise is large. The theoretical framework is applied to explain various phenomena in gradient based learning and to suggest new algorithms that enforce constraints.

**Strengths:**

Deriving properties of learning from symmetry in the loss function alone is original. This approach poses little assumption on the architecture, thus making the conclusions more general than previous studies.

Besides bringing a novel perspective, the new framework based on symmetry provides explanations to a wide range of phenomena in neural network training. Its ability to unify the theory behind these phenomena demonstrates the usefulness of the proposed approach.

The writing is clear, and each theorem is accompanied by helpful and intuitive explanations.

**Weaknesses:**

Section 2.1 mentions ReLU network as an example with rescaling symmetry. ReLU networks are not differentiable at some points in the parameter space. However, the proof of theorem 1 assumes that $\ell_0$ is twice differentiable, at least in the neighborhood of all points with $(u,w)=(0,0)$.

Depending on the geometry of the loss function, $\gamma$ that satisfies (21) may diverge close to (u,w)=(0,0). Hence, while Theorem 1 part 2 can be interpreted as locally favoring solutions (u,w)=(0,0), the required regularization strength can be too large to make this interpretation meaningful. Part 2 of theorem 2 and 3 have the same issue.

Some derivations in the main text skip steps, which makes it hard to follow for readers less familiar with related work. For example, it would be helpful to include some background information or reference for the escape rate of SGD in the third paragraph on page 7. In the following paragraph, the assumption that $H(x)$ commutes with $H(x’)$ might need a justification.

While the theory is successful in explaining various known phenomenon in training, there is limited evidence of its potential in guiding practices. Including some applications, such as experiments to back up the effectiveness of the algorithm in Section 4.6, could make the value of the theory more convincing.

**Questions:**

Lemma 2.1 in (Simsek 2021) also characterizes an attractive subspace derived from symmetry. Could the authors comment on whether this subspace is related to the stationary condition (Definition 3)?

Regarding footnote 5, would it be possible to prove that a twice differentiable function with bounded parameters cannot have a negatively diverging Hessian eigenvalue?

---

> ### Author Response · Authors · 2023-11-20
> **Reply part 1**
>
> Thank you for the detailed feedback. We answer both the weaknesses and questions below.
>
> Weaknesses:
>
> **1. Section 2.1 mentions ReLU network as an example with rescaling symmetry. ReLU networks are not differentiable at some points in the parameter space. However, the proof of theorem 1 assumes that $\ell_0$ is twice differentiable, at least in the neighborhood of all points with $(u,w)=(0,0)$.**
>
> This is true. As is clear from the requirement that the loss function needs to be differentiable, the theorems do not apply to the rescaling symmetry due to the ReLU activation. We have made this point clear in the manuscript.
>
>
> **2. Depending on the geometry of the loss function, $\gamma$ that satisfies (21) may diverge close to (u,w)=(0,0). Hence, while Theorem 1 part 2 can be interpreted as locally favoring solutions (u,w)=(0,0), the required regularization strength can be too large to make this interpretation meaningful. Part 2 of theorem 2 and 3 have the same issue.**
>
> Thanks for this criticism. We acknowledge that part 2 of these theorems is qualitative in nature, and giving a quantitative estimate of the required $\gamma$ remains an important open problem and can be highly problem-dependent. However, please let us explain why this is not really a weakness but a feature of the theory.
>
> First of all, we would like to point out that there is no divergence close to (u,w) = (0,0). The divergence is only possible if the loss function is not differentiable at (0,0), which is not possible by the assumption of twice-differentiability.
>
> Secondly, we note that it is possible to construct a dataset such that the required $\gamma$ is arbitrarily large -- however, this is not a problem but a feature. The magnitude of the required $\gamma$ is in general proportional to the strength of the correlation in the data. The best example to illustrate this problem is the simple 1d regression problem with rescaling symmetry: $L=(uwx-y)^2 + \gamma(u^2 + w^2)$. The minimizers of this objective are identical to the classical lasso objective (where $\gamma$ is equivalent to the regularization coefficient of the L1 penalty in lasso). At what $\gamma$ does $(0,0)$ become a sparse solution? The analytical solution is known: $\gamma=|xy|$. Namely, a sparse solution is preferred precisely when the regularization is stronger than the feature. By increasing $|xy|$, one can certainly make the required $\gamma$ arbitrarily large, but the same "problem" exists in lasso -- the stronger the data correlation, the larger the required L1 penalty to get a sparse solution. Therefore, this is not really a problem of the regularization method -- it merely reflects the (desired) fact that a strong regularization is required to prevent an easy overfitting.
>
> Thirdly, we would like to point out that, empirically, this is not really a problem for common deep learning problems. The reason is that the required $\gamma$ to cause a collapse in a certain direction can be approximated with the smallest eigenvalue of the Hessian in this direction. In deep learning, empirical works have found that the majority of the eigenvalues of the Hessian of a network is close to zero, with only a few large outliers (for example, see https://arxiv.org/abs/1706.04454). Therefore, for the majority of the direction to collapse/become sparse, only a small $\gamma$ is required. This is also consistent with our experiments in Figure 1 and Appendix A.

---

> ### Author Response · Authors · 2023-11-20
> **Reply part 2**
>
> **3. Some derivations in the main text skip steps, which makes it hard to follow for readers less familiar with related work. For example, it would be helpful to include some background information or reference for the escape rate of SGD in the third paragraph on page 7. In the following paragraph, the assumption that $H(x)$ commutes with $H(x')$ might need a justification.**
>
> Thanks for this criticism. When the dynamics follows SGD with minibatch noise, the local dynamics (and attractivity) is characterized by the Lyapunov exponent of the random matrix product process. The characterization of the Lyapunov exponent for this problem is a well-known open problem (see below). We have updated the manuscript to point to the study of escape rates for these problems.
>
> Regarding the commutation approximation, we use this approximation to give a qualitative characterization of the Lyapunov exponent and highlight its nonlinear dependence on the learning rate, batch size, and gradient distribution. This approximation suffices for our qualitative discussion. That being said, it should be noted that it is not a very accurate approximation, and the computation and estimation of the Lyapunov exponent for this process is, in fact, a well-known open problem for the field of dynamical systems and ergodic theory. For example, see Maximal Lyapunov exponents for random matrix products | SpringerLink for its discussion and advanced methods for estimating the Lyapunov exponents. We have also added these parts to the manuscript to help clarify this point to the readers.
>
>
>
> **4. While the theory is successful in explaining various known phenomenon in training, there is limited evidence of its potential in guiding practices. Including some applications, such as experiments to back up the effectiveness of the algorithm in Section 4.6, could make the value of the theory more convincing.**
>
> Thanks for this suggestion. We have now added an experiment to compare the learning dynamics of a vanilla ResNet18 and ResNet18 with the rescaling symmetry (a special case of the proposed DCS algorithm). The experiment shows that there is no significant difference between the final performance and the training time between the two, while the ResNet18 with the rescaling symmetry is sparse overall. See Section A4.
>
>
> Questions:
>
> **1. Lemma 2.1 in (Simsek 2021) also characterizes an attractive subspace derived from symmetry. Could the authors comment on whether this subspace is related to the stationary condition (Definition 3)?**
>
> Thanks for pointing to this relevant lemma from Simsek et al. This lemma characterizes the stationarity of the symmetry subspace due to the permutation symmetry when training proceeds with gradient flow. In agreement with our result, the symmetry subspace identified in Simsek agrees with our identification of the stationary condition for the permutation. The difference is that, certainly, our theory deals with discrete-time SGD under the minibatch stochasticity. We have added a discussion of this point to the manuscript.
>
> **2. Regarding footnote 5, would it be possible to prove that a twice differentiable function with bounded parameters cannot have a negatively diverging Hessian eigenvalue?**
>
> Yes. By the definition of the $C_2$ function class, we know that the second partial derivatives of the loss function are continuous. When restricted to a closed and bounded domain, a continuous function is bounded. Therefore, the Hessian matrix is always bounded, which implies that its eigenvalues are also bounded.

---

> > ### Comment · Reviewer_Wp9T · 2023-11-23
> >
> > Thank you for your response. I appreciate the added clarifications and the new experiment that compares learning dynamics with and without the rescaling symmetry. However, while I agree that the qualitative nature of the part 2 of theorem 1-3 is a feature of these theorems, I am not convinced that this does not affect the significance of the result. Therefore, I am inclined to maintain my score.

---

> > > ### Author Response · Authors · 2023-11-23
> > > **Author Reply 2**
> > >
> > > Thanks for your criticism. However, we are afraid to say that we cannot agree that this criticism shows that our result has limited significance. Please let us explain why.
> > >
> > > **1. However, while I agree that the qualitative nature of the part 2 of theorem 1-3 is a feature of these theorems, I am not convinced that this does not affect the significance of the result.**
> > >
> > > While we agree that the criticism is valid, we find it difficult to agree that it affects the significance of the result. The reviewer's main criticism is that $\gamma$ can be arbitrarily large, but this is a generic criticism that applies to any regularization method (in deep learning or in conventional statistical learning), as we explained in our first round of rebuttal.
> > >
> > > Let us quote the original criticism here: "**Depending on the geometry of the loss function, $\gamma$ that satisfies (21) may diverge ... the required regularization strength can be too large to make this interpretation meaningful.**" There are four major problems with this criticism:
> > > 1. it is generic because it can be applied to any regularization method. For example, Lasso achieves sparsity at a high regularization strength. One can also criticize by saying: "Depending on the problem, the required regularization strength may diverge, and so lasso does not meaningfully lead to sparsity." But this is not really a valid attack on the significance of Lasso because the value of the regularization strength is supposed to be highly problem/data-dependent
> > > 2. this is an impossible requirement. The required value of $\gamma$ is highly problem-dependent, depending on both the data distribution and architecture. Without specifying the problem, one cannot make such a computation or even suggest a value for $\gamma$. As an example, in the lasso problem, it is meaningless to suggest a hyperparameter without knowing the data. The fact that there is no quantitative characterization here reflects the generality of our result, not its limitation
> > > 3. it is **not** difficult to compute the critical value of $\gamma$ once we specify a problem, but computing it does not constructively add to our result. The required value of $\gamma$ is not difficult to compute in general. One good example is the problem of probabilistic PCA (see https://www.jstor.org/stable/2680726). This problem has a rotation symmetry between the encoder and decoder, and so our theory applies. At a large prior strength (equivalent to the regularization term in our theory), the solution becomes low rank. The critical values of the regularization strength can be analytically computed -- they are just the singular values of the data covariance matrix
> > > 4. this criticism ignores all the empirical results we provided. The point of our experiments is to show that the required $\gamma$ is within the standard range in common deep-learning settings. For example, see the ResNet18 experiment. The homogeneity effect is significant at the most commonly used values of weight decay. Why is this not meaningful?
> > >
> > > Also, there seems to be some (perhaps minor) misunderstanding of our result. Part 2 of Theorems 1, 2, and 3 are just corollaries of part 3 of Theorem 4. We find it confusing to criticize part 2 of Theorems 1, 2, and 3 but not part 3 and part 4 of Theorem 4.

---

### Official Review · Reviewer_HdbE · 2023-11-01

**Soundness:** 4 excellent
**Presentation:** 3 good
**Contribution:** 3 good
**Rating:** 8
**Confidence:** 2

**Summary:**

A novel theory of neural network symmetries is put forward which is loss function and architecture agnostic. Namely, a class of mirror symmetries are defined which includes rescaling, permutation, and rotation symmetries, and these mirror symmetries are shown to interact with weight decay to create absorbing states that cause network training to preferentially converge to certain low-rank solutions. The implications of this theory on empirical phenomena, including block structure Hessians, SGD convergence to saddle points, neural collapse, and regularization, are discussed.

**Strengths:**

The paper covers a lot of ground and generally presents its contents clearly and in a well-organized fashion. It gives an elegant and very intriguing perspective on loss landscapes through symmetry alone. It also unifies many different symmetries under the common language of $O$-symmetries, which looks to be a powerful mathematical perspective. The theory disentangles the effects of symmetries from weight decay and other regularization techniques on the loss landscape, which could enable new avenues of attack on controlling the minima that training converges to.

**Weaknesses:**

My main concern is that the relevance of the work could be made explicit, as at first glance it seems to restate obvious results (such as that zeroed weights remain zero throughout training). In particular it would be nice to get a summary of how changes to weight decay, learning rate, and the amount of mirror symmetries affect empirical outcomes, such as more weight decay = more likely to fall into symmetry-induced stationary conditions.

In section 4.4, I don't believe this theory should take credit for the solution of adding random gradient noise, since it seems a trivial way to eliminate all stationary conditions regardless of whether they are caused by symmetry or not (the random bias method is more relevant). In general the connections in 4.4 require more work and empirical evidence in order to understand which are caused by symmetry and which are not. Overall, the main implications of this paper could be hammered out more strongly, while less central ideas like relating collapse to "phase transitions" and the DCS algorithm could be cut or left as a mention in the discussion.

Minor problems:
- Some details/notations could be explained more thoroughly, such as: define "structured constraint" when it is introduced, what is $R$ referring to in (3), same symbol $O$ is used for both mirror and order in (8), definition of $\lambda$ versus $\eta$ in (9).
- The statement "Notably, Entezari et al. (2021) empirically showed that neural networks under SGD
likely converge to the same type of solution if we take permutation symmetry into consideration" isn't quite right, as Entezari conjectures this and follow-up work Ainsworth et al. shows something weaker (linear connectivity between pairs of solutions).

**Questions:**

Adding a fixed random bias to all parameters clearly prevents $O$-symmetries, but would there still be symmetries present? E.g. if two weights could previously be mirrored about a hyperplane through the origin, after adding the bias they can now be mirrored about a parallel hyperplane that is offset by the bias vector. How does this method eliminate the structured constraints such as in figure 3?

---

> ### Author Response · Authors · 2023-11-20
> **Reply**
>
> Thanks for the detailed feedback and the encouraging review. We answer both the weaknesses and questions below.
>
> **1. My main concern is that the relevance of the work could be made explicit, as at first glance it seems to restate obvious results (such as that zeroed weights remain zero throughout training). In particular it would be nice to get a summary of how changes to weight decay, learning rate, and the amount of mirror symmetries affect empirical outcomes, such as more weight decay = more likely to fall into symmetry-induced stationary conditions.**
>
> Thanks for this suggestion. We have updated the introduction to explicitly discuss the contributions early in the manuscript. We included more discussion of previous works in the introduction to clarify the motivation of our work from the beginning. In particular, prior to our work, symmetry has been studied in a case-by-case manner. Our work is the first to study symmetry through a unified framework. Second, previous works mostly take the perspective that symmetry creates saddle points and thus hinders training. In contrast, our current work considers how symmetry has a regularization effect and leads to structures and constraints of learning. We also include more detail when describing our contribution.
>
> **2. In section 4.4, I don't believe this theory should take credit for the solution of adding random gradient noise, since it seems a trivial way to eliminate all stationary conditions regardless of whether they are caused by symmetry or not (the random bias method is more relevant). In general the connections in 4.4 require more work and empirical evidence in order to understand which are caused by symmetry and which are not. **
>
> Thanks for this comment. It is not our intention to take credit for adding random gradient noise to the training, as it has been used in deep learning practice and proposed by previous authors in various situations. In the context of the loss of plasticity problem, it has also been proposed by previous authors as a solution. We have added a reference and clarified this point in the revision.
>
> Minor problems:
>
> **1. Some details/notations could be explained more thoroughly, such as: define "structured constraint" when it is introduced, what is R referring to in (3), same symbol O  is used for both mirror and order in (8), definition of $\lambda$ versus $\eta$ in (9).**
>
> Thanks for pointing out these problems. We have removed the inconsistent notations and clarified the meanings of these words and notations in the updated manuscript. (3): $R$ should be $\Omega$. (8): We now avoid using $O$ to denote order. (9): $\eta$ should be $\lambda$.
>
>
> **2. The statement "Notably, Entezari et al. (2021) empirically showed that neural networks under SGD likely converge to the same type of solution if we take permutation symmetry into consideration" isn't quite right, as Entezari conjectures this and follow-up work Ainsworth et al. shows something weaker (linear connectivity between pairs of solutions)**
>
> Thanks for pointing this out. We intended to say that there exists various empirical evidence supporting the conjecture that SGD converges to the same type of solution if we take permutation symmetry into consideration. We have removed this part from the manuscript as it is not an essential discussion for us.
>
> Question:
>
> **1. Adding a fixed random bias to all parameters clearly prevents O-symmetries, but would there still be symmetries present? E.g. if two weights could previously be mirrored about a hyperplane through the origin, after adding the bias they can now be mirrored about a parallel hyperplane that is offset by the bias vector. How does this method eliminate the structured constraints such as in figure 3?**
>
> This is a very good question. In short, it is usually not symmetries that lead to bad solutions but the fact that these symmetries tend to appear together with small-norm solutions that are low-capacity. This is especially true in the case of $O$-mirror symmetries, which pass through the origin. However, when the symmetry does not pass through the origin, the symmetric solutions often do not coincide with small-norm solutions (which are preferred when there is weight decay), and are thus more likely to be benign. To give an example, consider the problem $L=(uwx -y)^2$. Here, the symmetry solution is $u=w=0$, which is also low-capacity. When we add a bias, say $L=((u-0.1)(w-0.1)x -y)^2$,  the situation is different. The symmetric solution no longer coincides with a small-norm solution, and in this case, adding weight decay actually prevents convergence to the symmetric solution and is helpful for optimization. We have added this discussion to the manuscript.

---

### Official Review · Reviewer_Z5Gq · 2023-11-02

**Soundness:** 3 good
**Presentation:** 1 poor
**Contribution:** 2 fair
**Rating:** 5
**Confidence:** 3

**Summary:**

The paper considers a connection between the symmetries of the weights of a neural network and the constraints of the loss function. The authors proof that mirros symmetries lead to pre-defined certain behaviours of gradient-based model learning. For instance, rescaling symmetry of weights, leads to sparsity, rotation - to low-rankness and permutation leads to ensembling.

While the paper presents 4 theorems with possibly useful application, the flow of the paper, the content, require a significant revision.

#### Final Decision
After reading the other reviews and the answers, I stick to my rating

**Strengths:**

- The authors present four theorems. Each theorem is well-written on its own, and the proofs are correct, with a clear and easy-to-follow flow.
- The authors show that the paper has numerous connections to various fields in machine learning, which potentially opens up many possibilities for practical applications.

**Weaknesses:**

The main weakness of the paper is that the text itself does not allow the reader to immediately position the contributions of the paper in the realm of machine learning. It is not clearly stated what alternatives to this approach exist.

- The motivation of the paper is not clearly stated in the very beginning of the paper. The importance of the contribution becomes more and more clear only to the very end.
- There is no "Related work" section. The way it is presented in Section 4 does not sound coherent and does not allow the reader to properly position the work.
- The overall flow of the paper is not smooth. It jumps back and forth, there is no structure, there is no clear finale.
- There is no clear section with experiments. Figure 1 demonstrates some results. However, the results seem more like an illustration rather than a solid experiment. It should be built as an experiment with its motivation, details, and explanation. And it should be in the main part of the paper.
- Overall, the experiments do no convince.

**Questions:**

- In Eq 11, you suggest a reparametrization where $\theta$ is replaced with $u, v, w$. Does it mean that the total number of trainable parameters will increase by a factor of 3?
- Can you elaborate on the dynamics with which the neural network is sparsified if a scaling symmetry exists? Do you have any estimates on how long it will take for a weight to become less than $\epsilon \ll 1$? I can imagine, that in practice, due to the stochastic nature of optimization, the mechanism will work differently.

---

> ### Author Response · Authors · 2023-11-20
> **Reply part 1**
>
> Thank you for the detailed feedback. We answer both the weaknesses and questions below.
>
> Weaknesses:
>
> **1. The motivation of the paper is not clearly stated in the very beginning of the paper. The importance of the contribution becomes more and more clear only to the very end.**
>
> We have updated the introduction in great detail to clarify the contribution and related works from the beginning of the work. We have significantly revised the introduction to clarify the motivation and background of our work. In particular, prior to our work, symmetry has been studied case-by-case. Our work is the first to study symmetry through a unified framework. Second, previous works mostly take the perspective that symmetry creates saddle points and thus hinders training. In contrast, our current work considers how symmetry has a regularization effect and leads to structures and constraints of learning.
>
> **2. There is no "Related work" section. The way it is presented in Section 4 does not sound coherent and does not allow the reader to properly position the work.**
>
> As we answered in weakness 1, we have updated the introduction to discuss the related works more extensively from the beginning. We also updated the main text overall to remove incoherent wording carefully. That being said, if you give more specific references regarding which part of the discussion in section 4 is not sufficiently coherent, we are happy to revise that part as well.
>
> **3. The overall flow of the paper is not smooth. It jumps back and forth, there is no structure, there is no clear finale.**
>
> Thanks for this criticism. We have modified the last paragraph in the introduction to delineate the structure of the paper clearly. In short, our structure flows as follows: introduction -> three specific examples -> general theory -> implications. In terms of the finale, we have made some minor revisions to the conclusion to condense the points we want to make. If you think there is any specific aspect of the conclusion that would be improved, we are happy to hear and make further revision.
>
> **4. There is no clear section with experiments. Figure 1 demonstrates some results. However, the results seem more like an illustration rather than a solid experiment. It should be built as an experiment with its motivation, details, and explanation. And it should be in the main part of the paper.**
>
> Thanks for the criticism. First of all, we would like to point out that the main contribution of our work is theoretical and conceptual, and these experiments are indeed intended to be illustrations of the theoretical results.
> Secondly, an illustrative experiment can be solid at the same time, and we believe that our experimental results are solid. Due to space constraints, the experimental details are all presented in Section A2 with great technical detail and explanation. What we have included in the main texts are results and summaries from these experimental analyses.
> That being said, to improve the comprehensibility of these experimental results, we have included more references and explanations of the experiments in the main text. See the newly added Section 4.7 in the revision.
>
> **5. Overall, the experiments do no convince.**
>
> With the added explanation and experimental details we added to answer point 4, we believe that the experiments are now convincing. If you think any specific point is unclear, please let us know so that we can provide more detail.

---

> ### Author Response · Authors · 2023-11-20
> **Reply part 2**
>
> Questions:
>
> **1. In Eq 11, you suggest a reparametrization where $\theta$  is replaced with $u, w, v$. Does it mean that the total number of trainable parameters will increase by a factor of 3?**
>
> Yes, but not necessarily. The factor of 3 is an upper bound, which is only required when one wants a complicated constraint. In more common situations, one can greatly reduce the number of parameters required. For example, one only needs to add a single parameter to introduce a rescaling symmetry, which equivalently adds a simple sparsity constraint to one parameter. To introduce simple sparsity constraints to all the parameters, one only needs to double the amount of the parameters. Another good example is group sparsity, where one only needs to add one single parameter for each group. Normally, the number of sparse groups is equal to the number of input features (such as in a feature selection problem), and so the number of additional parameters (of the order of hundreds or thousands) is far less than the total number of parameters (of the order of a million and above).
>
> Also, we would like to comment that multiplying the training parameters by a factor of 2 or three is usually not too much of a problem for modern GPU-based training. This is because when the training proceeds with minibatch training, the predominant cost of memory and computation comes from the dynamic cost of computation over the minibatch, not from the static cost of increasing the number of parameters by a factor of two or three. See the newly added experiment in Section A4.
>
>
> **2. Can you elaborate on the dynamics with which the neural network is sparsified if a scaling symmetry exists? Do you have any estimates on how long it will take for a weight to become less than $\epsilon \ll 1$? I can imagine, that in practice, due to the stochastic nature of optimization, the mechanism will work differently.**
>
> This is in fact a good and important open question. One can obtain a rough estimate in simplest scenarios when training proceeds with gradient flow (GF). Consider optimizing a problem $L =(uvx -y)^2$ with GF -- this problem can be exactly solved, and the learning time scale can be found to be $1 / \eta x y$, where $\eta$ is the learning rate, and $xy$ can be identified with the strength of the signal in the training data. Therefore, the training loss decreases by one order after $1 / \eta x y$. This can be compared with the case when there is no rescaling symmetry: $L=(ux-y)^2$, where the relaxation time scale is $1/\eta x^2$. The two time scales are different, but neither is categorically better than the other. Roughly speaking, one would expect that learning with symmetry will not be significantly slower than without symmetry.
>
> We also performed an experiment to compare the learning dynamics of a vanilla ResNet18 and ResNet18 with the rescaling symmetry. The experiment shows that there is no significant difference between the final performance and the training time between the two, while the ResNet18 with the rescaling symmetry is sparse overall. See the newly added Section A4.

---

### Official Review · Reviewer_tSGN · 2023-11-23

**Soundness:** 3 good
**Presentation:** 3 good
**Contribution:** 3 good
**Rating:** 6
**Confidence:** 4

**Summary:**

The paper presents a framework that analyze the local and global geometry of neural network with L2 regularization, using the loss symmetries, without any knowledge about the underlying network's architecture. The paper deals with 4 types of symmetries (rescaling, rotation, permutation and mirroring) and describes how each symmetry affects the geometry of the loss landscape with respect to the regularization coefficient. The paper uses the framework to analyze 5 different real world phenomenons and implement one algorithm.

**Strengths:**

The paper presents a novel approach to understand how symmetries in the loss function of a neural network regardless of their underlying architecture. It enumerates multiple real world application to justify the technique's importance.

**Weaknesses:**

The paper is hard to read and has some mathematical inaccuracies, I'll enumerate some of problems I encountered:

(1) The proof of theorem 4: there is a recurring typo writing ker(O) instead of ker(O^T) and the RHS of eq. 52 should be multiplied by -1 for the proof to hold

(2) Assumptions 1 and 2 are given without any explanation. It isn't self-evident that these are reasonable assumptions for real-world scenarios.

(3) Chapter 2.2 uses both big omega and R as rotation matrices, but R wasn't declared. In addition, item 1 in theorem 2 doesn't says for which gammas the theorem holds (I assume from the context it is for every gamma).

(4) In the line below equation 9, eta is used for learning rate, but it looks like gamma instead in subsequent paragraphs.

**Questions:**

No questions.

---

> ### Author Response · Authors · 2023-11-23
> **Author reply**
>
> Thanks for the criticism. While this criticism is posted less than 2 hours before the rebuttal period ends, we do our best to make a revision and rebuttal. In summary, there is no mistake in the proof, and the assumptions we made are well-justified. We feel that the criticisms are perhaps due to misunderstanding and are only targeted at minor problems.
>
> **The paper is hard to read and has some mathematical inaccuracies**
>
> Thanks for the criticism. We have updated the manuscript to improve accuracy. We have also significantly updated the introduction to clarify our motivation and contribution from the beginning. In addition, if you could be more specific about why the paper is hard to read, we can revise that part accordingly.
>
> **(1) The proof of theorem 4: there is a recurring typo writing ker(O) instead of ker(O^T)**
>
> Thanks for pointing out. The typo has been fixed.
>
> **The RHS of eq. 52 should be multiplied by -1 for the proof to hold**
>
> We believe this is a misunderstanding of the proof. The RHS of Eq. (52) does not need to be multiplied by -1 for the proof to hold.  We have rewritten the proof of this part to improve its notational clarity and give all the details after Eq. (52) to clarify the steps.
>
> **(2) Assumptions 1 and 2 are given without any explanation. It isn't self-evident that these are reasonable assumptions for real-world scenarios.**
>
> First of all, we do give explanations of these assumptions in the immediate context.
>
> Let us focus on assumption 1, as it relates to our main contribution. Essentially, assumption 1 assumes that the Hessian of the loss function is lower bounded. This is a minimal smoothness assumption and not a strong assumption at all. It is satisfied by, for example, any convex function. The meaning is also quite clear. Having a negative eigenvalue in the Hessian means that the loss can be locally concave, and its magnitude can be regarded as a measure of "local concavity" -- in this language, the assumption basically assumes that the local concavity does not diverge -- which is a very weak condition.
>
> Mathematically, this is also not a problem because this assumption can be easily replaced by requiring the loss function to have a bounded domain. This is because any $C_2$ function on a closed and bounded domain must have a bounded Hessian, which implies that its eigenvalues are also bounded. Assuming that the parameters are constrained in a compact set is certainly benign, as well-trained large neural networks tend to stay close to their initialization.
>
> The fact that this assumption is benign is also supported by empirical evidence. Any existing work that empirically computes the Hessian spectrum has only found very small negative eigenvalues in the Hessian (not to mention a diverging negative Hessian eigenvalue). For a classical example, see https://arxiv.org/abs/1706.04454. Alternatively, within the regime of NTK, the loss function is essentially convex. In summary, for almost all practical concerns, assumption 1 is benign and should leads to no problem.
>
> Now, let us take a step back. Even if assumption 1 is problematic, it only affects item 4 of Theorem 4, which is the least important of our result. In fact, none of our major discussion really relies on item 4 of Theorem 4. We find it difficult to understand why this criticism justifies a soundness score of 1 and/or rejection.
>
> For assumption 2, we point out that assumption 2 only appears in the appendix, is not central to our main result, and only serves the purpose of discussing the relationship of our result with previous results. That being said, it rationale is well-explained in the context where it appears. See the discussion above and below Assumption 2.
>
> **(3) Chapter 2.2 uses both big omega and R as rotation matrices, but R wasn't declared.**
>
> Here, $\Omega$ is used for the rotation symmetry. $R$ is used for the double rotation symmetry. Also, we point out that $R$ is defined exactly in the sentence where it is introduced, which we quote here: "A more common symmetry is a ``double" rotation symmetry, where $\ell_0$ depends on two matrices $U$ and $W$ and satisfies $\ell_0(U, W) = \ell_0(UR, R^TW)$, for any orthogonal matrix $R$ and any $U$ and $W$."
>
> **In addition, item 1 in theorem 2 doesn't says for which gammas the theorem holds (I assume from the context it is for every gamma).**
>
> This is true. Item 1 holds for any $\gamma$ (even for negative $\gamma$), and so there is no need to state this.
>
>
> **(4) In the line below equation 9, eta is used for learning rate, but it looks like gamma instead in subsequent paragraphs.**
>
> Thanks for pointing this out. We have fixed this typo.

---

### Author Response · Authors · 2023-11-20
**Rebuttal summary**

First of all, we would like to thank all the reviewers for their careful and constructive feedback. We have carefully studied the criticism and feedback from the referees and adapted our manuscript accordingly to address the raised concerns. The updated parts are highlighted in orange.

After the revision, we are confident that the manuscript is significantly improved and the actual contributions are greatly clarified. To summarize, the following are the major revisions we made to the manuscript:

1. The introduction is rewritten to clarify the motivation and contribution of our work from the beginning
2. More discussion is added regarding the Lyapunov exponent, especially regarding the commutation approximation and our current understanding of it (footnote 6)
3. The gradient noise injection method is attributed to proper prior works (footnote 8)
4. A key difference between $O$-symmetries and symmetries with respect to hyperplanes that do not pass through the origin is clarified (footnote 9)
5. The experimental results are now discussed and explained in the main text (section 4.7)
6. A new experiment that compares the learning dynamics of vanilla ResNet18 and ResNet18 with rescaling symmetry is presented in Section A4. Here, it is clear that rescaling symmetry leads to parameter sparsity, in agreement with the theory

Below, we answer the criticisms and questions of each reviewer in detail.

---

### Meta-Review · Area_Chair_5fUj · 2023-12-05

**Metareview:**

The paper is aimed at contributing to existing large body of work studying different ways by which invariance of the loss function to certain groups restricts the loss landscape. Here, emphasis is placed on regularized losses invariant to certain Householder reflections, referred to as $O$-mirror symmetry.

The reviewers appreciated the proposed use of the $O$-mirror symmetry, but generally thought that the motivation to the work as well as its relevance to practice should be made more solid. In addition, some have raised several issues concerning the presentation of the results, in particular finding it hard to asses their significance based on pointers provided to related work. The authors are encouraged to incorporate the important feedback given by the knowledgeable reviewers.

**Justification For Why Not Higher Score:**

A substantial revision is needed before the paper can be recommended for acceptance.

**Justification For Why Not Lower Score:**

N/A

---

### Decision · Program_Chairs · 2024-01-16

Reject